# Etv transcription factors functionally diverge from their upstream FGF signaling in lens development

**Ankur Garg**[1,2†‡], **Abdul Hannan**[1,2†], **Qian Wang**[1,2†], **Neoklis Makrides**[1,2], **Jian Zhong**[3], **Hongge Li**[1,2§], **Sungtae Yoon**[1,2], **Yingyu Mao**[1,2], **Xin Zhang**[1,2*]

[1]Department of Ophthalmology, Columbia University, New York, United States; [2]Department of Pathology and Cell Biology, Columbia University, New York, United States; [3]Burke Neurological Institute and Feil Family Brain and Mind Research Institute, Weill Cornell Medicine, White Plains, United States

**Abstract** The signal regulated transcription factors (SRTFs) control the ultimate transcriptional output of signaling pathways. Here, we examined a family of FGF-induced SRTFs – *Etv1*, *Etv 4*, and *Etv 5* – in murine lens development. Contrary to FGF receptor mutants that displayed loss of ERK signaling and defective cell differentiation, *Etv* deficiency augmented ERK phosphorylation without disrupting the normal lens fiber gene expression. Instead, the transitional zone for lens differentiation was shifted anteriorly as a result of reduced Jag1-Notch signaling. We also showed that Etv proteins suppresses mTOR activity by promoting *Tsc2* expression, which is necessary for the nuclei clearance in mature lens. These results revealed the functional divergence between Etv and FGF in lens development, demonstrating that these SRTFs can operate outside the confine of their upstream signaling.

**\*For correspondence:**
xz2369@columbia.edu

[†]These authors contributed equally to this work

**Present address:** [‡]Department of Physiology, Cardiovascular research Institute, University of California, San Francisco, San Francisco, United States; [§]Department of Microbiology & Immunology, Indiana University School of Medicine, Indianapolis, United States

**Competing interests:** The authors declare that no competing interests exist.

## Introduction

The cell signaling networks are commonly depicted in a hierarchical manner, starting with the binding of extracellular ligands to cell surface receptors, followed by the relay of cytoplasmic mediators, and culminating in the activation of nuclear transcription factors. In this unidirectional view, each signal-regulated transcription factor (SRTF) is expected to control a subset of the transcriptional output of upstream signaling (*Cvekl and Zhang, 2017*). This model has been confirmed in many systems. For example, the transcriptomic changes induced by BMP, Hedgehog and Wnt signaling can be readily accounted for by their transcription effectors Smad, Gli and β-catenin, respectively. However, whether this principle applies to the SRTFs for FGF signaling remains to be determined.

The 28 mammalian E26 transformation-specific (ETS) proteins share a highly conserved winged helix-turn-helix DNA binding domain, which recognizes a core GGA sequence motif (*Charlot et al., 2010*). Post-translational modifications of these factors, especially serine/threonine phosphorylation by ERK, directly affect their subcellular localization, DNA binding and transactivation. In particular, FGF-ERK signaling induces expression of the *Etv* (*Pea3*) subfamily of ETS transcription factors, *Etv1* (*Er81*), *Etv4* (*Pea3*) and *Etv5* (*Erm*), in addition to enhancing their transcriptional activities during embryonic development (*Münchberg and Steinbeisser, 1999*; *Raible and Brand, 2001*; *Roehl and Nüsslein-Volhard, 2001*). Conditional knockouts of these *Etv* genes disrupted the anterior-posterior patterning of the limb bud and branching morphogenesis of the lacrimal gland controlled by FGF signaling (*Garg et al., 2018*; *Zhang et al., 2009*). The significance of *Etv* factors have also been demonstrated in studies of human cancer, where aberrant activation of *ETV* genes has been proposed to emulate oncogenic *RAS* in cellular transformation (*Hollenhorst et al., 2011*). Thus, the *Etv*

**eLife digest** Many cells contain proteins known as signal-induced transcription factors, which are poised to receive messages from the environment and then react by activating genes required for the cell to respond appropriately. It is commonly thought that these transcription factors faithfully follow the instructions they receive from the external signal: for instance, if the message was to encourage the cell to grow, the transcription factors would switch on growth-related genes.

As the eyes of mice and other mammals develop, a signal known as FGF is required for certain cells to specialize into lens fiber cells: these long, thin, transparent cells form the bulk of the lens, the structure that allows focused vision. Previous studies suggest that FGF activates three transcription factors known as Etv1, Etv4 and Etv5, but their precise roles in the development of the lens has remained unclear.

Here, Garg, Hannan, Wang et al. confirm that FGF signaling does indeed activate all three proteins. However, mutant mice that lacked Etv1, Etv4 and Etv5 still created lens fiber cells, suggesting that the transcription factors are largely unnecessary for lens fiber cells formation.

Instead, the Etv proteins participated in a cascade of molecular events involving a protein called Notch; as a result, if the transcription factors were absent, the lens fiber cells formed prematurely. In addition, deactivating Etv1, Etv4 and Etv5 also promoted the activity of a protein which interfered with the removal of internal cell compartments, a process required for lens fiber cells to mature properly. These findings reveal that the roles of Etv1, Etv4 and Etv5 deviate from and even oppose FGF signaling in the lenses of mice.

Transcription factors control the ultimate fate of a cell, and there is therefore increased interest in targeting them for therapy. The work by Garg, Hannan, Wang et al. reveals an unexpected complexity in how these proteins respond to upstream signals, highlighting the importance of further dissecting these relationships.

---

transcription factors *Etv1*, *Etv4* and *Etv5* are considered to be SRTFs that are directly downstream of FGF-Ras-ERK signaling.

FGF signaling is required during several steps of vertebrate lens development, including induction of the lens vesicle, proliferation of lens epithelial cells, and differentiation of lens fiber cells (*Cvekl and Zhang, 2017*; *Faber et al., 2001*; *Robinson, 2006*). The mature lens consists of a single layer of epithelial cells in the anterior and differentiated lens fiber cells in the posterior; the latter accounts for the bulk of the lens tissue. FGF signaling has been proposed to act in a gradient fashion, promoting proliferation in the lens epithelium at low signaling strength and stimulating differentiation in the lens fiber cells at high strength (*Lovicu and McAvoy, 2001*; *McAvoy and Chamberlain, 1989*; *McAvoy et al., 1991*). In support of this model, genetic knockouts of FGF receptors disrupted the expression of lens-specific genes *Maf*, *Prox1*, *Pax6*, *Cdh1* and *Crystallins*, affecting survival and proliferation of lens epithelial cells and elongation of fiber cells (*Chow et al., 1995*; *Collins et al., 2018*; *Garcia et al., 2005*; *Robinson et al., 1995a*; *Zhao et al., 2008*). Apart from FGF receptors, their co-receptors heparan sulphates and downstream mediators Frs2, Shp2, and Crk have also been demonstrated to regulate lens cell differentiation and elongation (*Collins et al., 2018*; *Li et al., 2019*; *Li et al., 2014*; *Madakashira et al., 2012*; *Pan et al., 2006*; *Qu et al., 2011*). The knockout phenotypes of these genes were reproduced by overexpression of negative regulators of FGF-ERK pathway, *Sef* and *Sprouty* (*Newitt et al., 2010*; *Shin et al., 2015*). On the other hand, transgenic overexpression of *Fgf1* or *Fgf3* resulted in premature differentiation of lens epithelial cells (*Collins et al., 2018*; *Robinson et al., 1998*; *Robinson et al., 1995b*), whereas over-activation of FGF signaling as a result of *Nf1* and *Spry1/2* deletion disrupted lens induction and lens fiber cell differentiation, respectively (*Carbe and Zhang, 2011*; *Kuracha et al., 2011*). These results demonstrated that the FGF signaling cascade is critical for lens development, but the direct downstream transcriptional effectors of FGF signaling were not well understood.

In this study, we investigated the role of *Etv* family transcription factors in the lens by genetically ablating *Etv1*, *Etv4,* and *Etv5*. Instead of the delayed cell differentiation phenotype expected from FGF signaling deficiency, we observed that the lens epithelial cells differentiated prematurely as a result of reduced Notch signaling. On the other hand, the expression of FGF targets *Maf* and

*Crystallins* were largely preserved in *Etv1/4/5* mutant lenses. We also showed that mTOR signaling was aberrantly upregulated, resulting in the failure of nuclei removal in the mature lenses. These results revealed the critical differences between the function of Etv family transcription factors and FGF signaling during lens development, demonstrating that these SRTFs can operate outside the confine of upstream signaling.

## Results

### MAPK-regulated Etv transcription factors are required for lens development

Previous studies have shown that Etv transcription factors are controlled by FGF signaling during embryonic development. To confirm this finding in the lens, we generated conditional knockouts of *Fgfr1* and *Fgfr2* using a lens-specific Cre driver *Pax6^{Le}-Cre*, also known as *Le-Cre*, which is active in the lens ectoderm as early as E9.5 (*Ashery-Padan et al., 2000*). As we and others have previously reported, genetic ablation of FGF receptors prevented the lens ectoderm from forming the lens vesicle in the *Pax6^{Le}-Cre; Fgfr1^{fl/fl}; Fgfr2^{fl/fl}* (*Fgfr* CKO) embryo (*Figure 1A–H*, arrows) (*Collins et al., 2018*; *Garcia et al., 2005*). Importantly, both ERK phosphorylation and expression of *Etv1*, *Etv4* and *Etv5* were also lost in the *Fgfr* CKO lens ectoderm (*Figure 1A–H*, outlined), demonstrating that FGF signaling indeed controls ERK and Etv activities during lens induction. We next ablated *Mek* and *Erk* to investigate whether *Etv* genes were also regulated by MAPK signaling in the lens. Although the lenses were formed in *Pax6^{Le}-Cre; Mek1(Map2k1)^{fl/fl}; Mek2(Map2k2)^{-/-}* (*Mek* CKO) and *Pax6^{Le}-Cre; Erk1(Mapk3)^{-/-}; Erk2(Mapk1)^{fl/fl}* (*Erk* CKO) embryos, they failed to express any of the *Etv* genes (*Figure 1I–Q*, circled). These results demonstrated that the *Etv* family transcription factors are controlled by FGF-ERK signaling during lens development.

To investigate the function of *Etv* family transcription factors in the lens, we used the *Pax6^{Le}-Cre* transgenic mouse line to conditionally ablate *Etv1* and *Etv5* in an *Etv4*-null background. In *Pax6^{Le}-Cre; Etv1^{fl/fl}; Etv4^{-/-}; Etv5^{fl/fl}* embryos (*Etv* TKO), the lens induction and lens vesicle formation were unaffected, but the lens size was reduced at E14.5 as shown by Hematoxylin and Eosin staining (*Figure 2A and E*). Consistent with this, apoptosis increased in the epithelium and the transitional zone as indicated by TUNEL staining (*Figure 2B,F and I*, arrows). On the other hand, proliferation of the lens epithelial cells was reduced as measured by EdU incorporation assay (*Figure 2C,D and I*, arrowheads). In the control lens, the nascent fiber cells exited the cell cycle after the transitional zone, as shown by the diminution of Cyclin D1 expression and Edu labeling (*Figure 2C*, arrows). In contrast, the *Etv* TKO lens displayed ectopic expression of Cyclin D1 in the posterior lens (*Figure 2G*, arrow) and persistent EdU- and pHH3-positive cells in the fiber cell compartment (*Figure 2C,D,G and H*, inserts). These phenotypes showed that lens development was disrupted by loss of *Etv* transcription factors.

### Genetic ablation of *Etv* causes ectopic activation of ERK signaling

To identify the downstream targets of *Etv* genes, we performed RNA sequencing analysis to compare the transcriptional landscape of E14.5 control to that of *Etv* TKO mutant. Since *Etv* genes are predominantly expressed in the transitional zone at the lens equator, we isolated these cells by laser capture microdissection and extracted RNA for cDNA synthesis (*Figure 3A*). After amplification, the cDNA library was subjected to high-throughput sequencing to examine the global gene expression changes. Cluster and PCA analysis of the top 200 differentially expressed genes showed that the controls and mutants were well segregated across three biological replicates, demonstrating the consistency of the RNA sequencing data (*Figure 3B* and *Figure 3—figure supplement 1A*). Among the 834 genes showing statistically significant changes in expression (p<0.05), 288 and 283 genes were up- and down-regulated by at least two folds, respectively. Interestingly, we noticed that the remaining transcripts of three *Etv* genes were reduced in mutant samples, suggesting that *Etv* genes regulated their own expressions (*Figure 3C*). These results indicate that loss of *Etv* genes causes complex changes in the lens transcriptome.

We next focused on the impact of *Etv* deletion on FGF signaling. The RNA sequencing data indicated that the receptors for FGF signaling, *Fgfr1* and *Fgfr3*, were down-regulated in the *Etv* TKO mutant (*Figure 3C*), which were confirmed by RNA in situ hybridization (*Figure 3—figure*

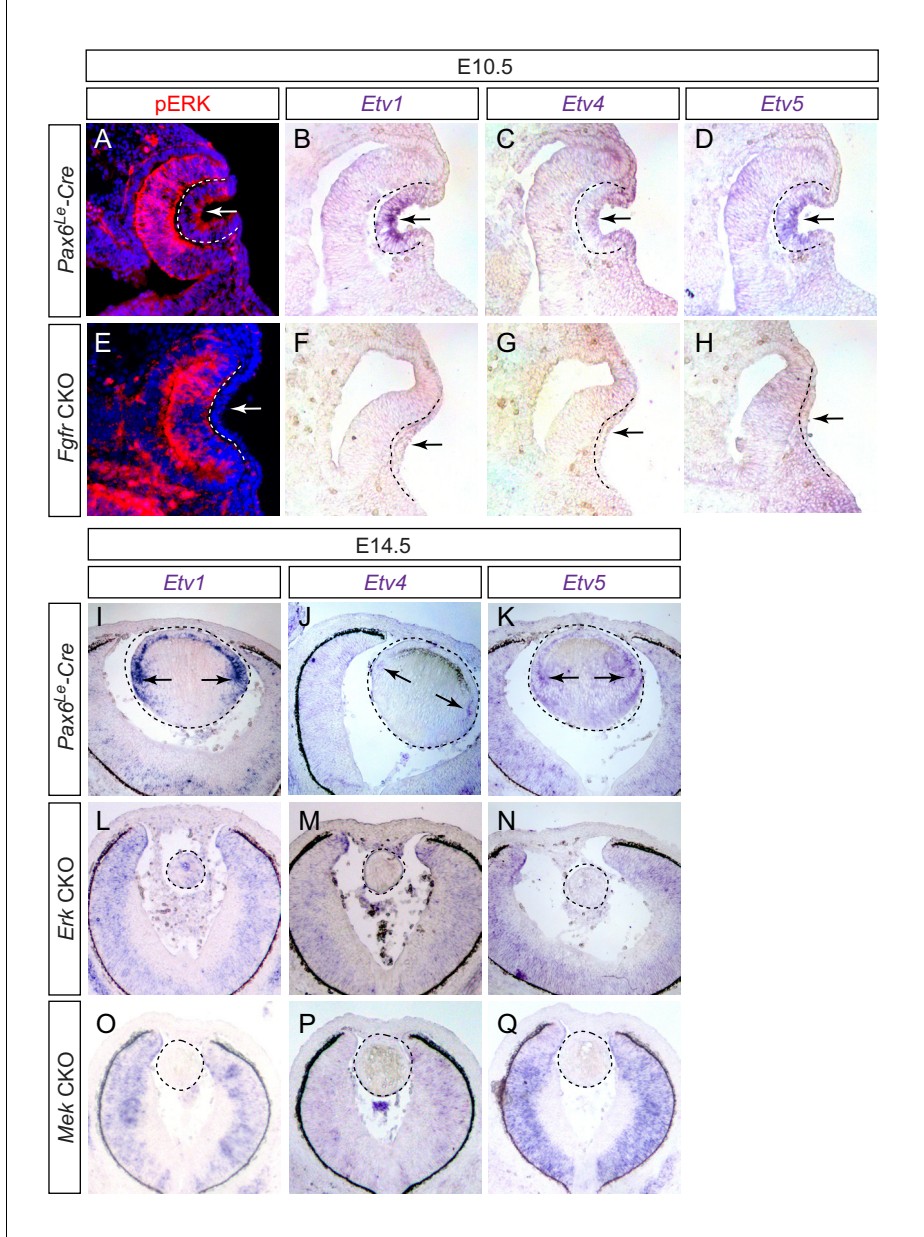

**Figure 1.** Etv transcription factors are controlled by FGF-ERK signaling in the lens. (A–D) At E10.5, the invaginating lens ectoderm displays ERK phosphorylation and expression of *Etv1*, *4* and *5* (arrows). (E–H) Genetic ablation of *Fgfr1/2* in *Fgfr* CKO mutants prevented lens vesicle formation and abrogated phospho-ERK and *Etv* expression (arrows). The lens ectoderms are marked by dotted lines. (I–K) At E14.5, *Etv1*, *4* and *5* are predominantly expressed in the transitional zone of the lens (arrows). (L–Q) Deletion of either *Erk1/2* (*Erk* CKO) or *Mek1/2* (*Mek* CKO) abolished expression of *Etv* genes. The lenses are circled in dotted lines.

*supplement 1B*). Although this suggests a positive feedback regulation of FGF receptors by *Etv* genes, it is unlikely to affect lens development, because previous studies have shown that *Fgfr1* and *Fgfr3* double mutant lenses did not exhibit any overt phenotype (*Zhao et al., 2008*). On the other hand, the transcript level of *Spry2*, an inhibitor of receptor tyrosine kinases, was also reduced, which we confirmed by RNA in situ hybridization in *Etv* TKO mutant lenses (*Figure 3D and F*, arrows). This is consistent with previous ChIP-seq analysis which revealed that *SPRY2* is a direct target of *PEA3/ ETV* family genes (*Yan et al., 2013*). Since we have shown above that expression of *Etv1*, *Etv4* and *Etv5* in the lens is dependent on ERK signaling (*Figure 1I–Q*), it is not surprising that *Spry2* expression is also lost in *Erk* CKO mutants (*Figure 3—figure supplement 1C*, arrows). Considering that

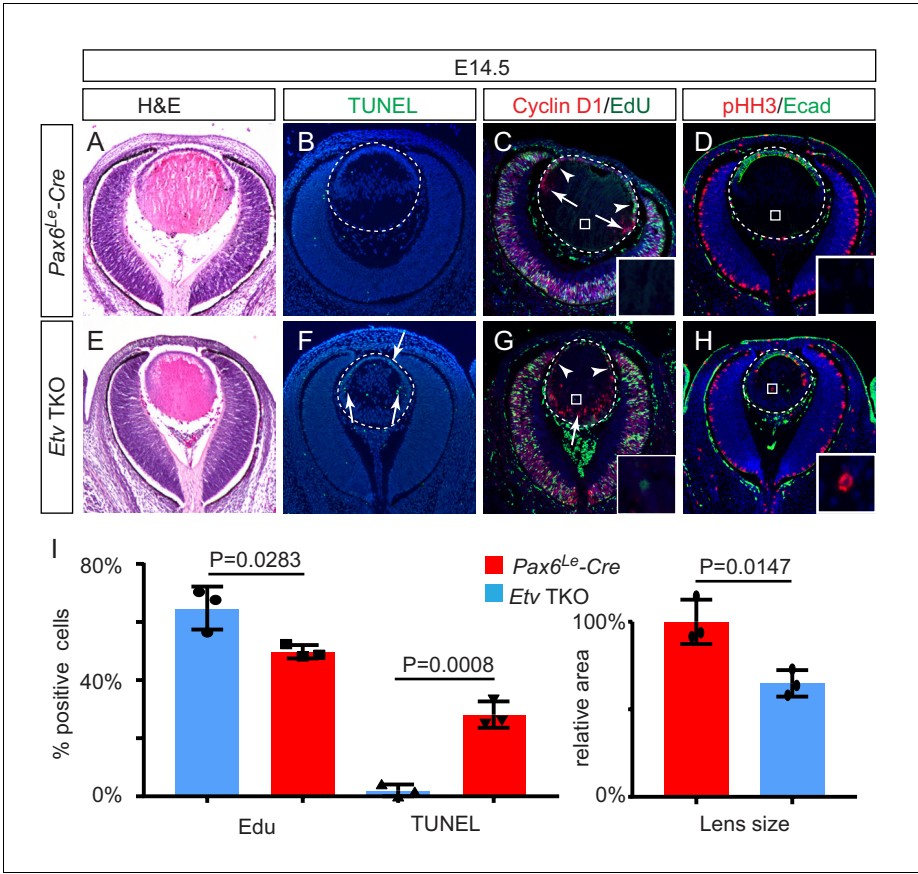

**Figure 2.** Lens development requires Etv transcription factors. (**A–H**) Hematoxylin and eosin (H and E) staining reveal reduced lens size in *Etv1/4/5* deletion (*Etv* TKO) mutants (**A and E**). *Etv* null lens exhibited increased cell apoptosis shown by TUNEL staining (B and F, arrows) and reduced cell proliferation as indicated by EdU staining (C and G, arrowheads). In contrast, ectopic expression of Cyclin D1 (G, arrows) and proliferation markers EdU and pHH3 (C, D, G and H, inserts) were detected in the posterior lens. (**I**) Quantification of EdU and TUNEL staining. The online version of this article includes the following source data for figure 2:

**Source data 1.** Source data for *Figure 2I*.

Sprouty proteins are known to inhibit Ras-MAPK signaling (*Hanafusa et al., 2002*), this suggests that *Etv* regulation of *Spry* constitutes a negative feedback loop to regulate FGF-ERK activity. Indeed, we notice that ERK phosphorylation was ectopically activated in *Etv TKO* mutant lenses (*Figure 3E and G*, arrows). In support of the upregulation of ERK activity, we also observed increasing expression of *Egr1* and *Fos* (*Figure 3C*), two early response genes for Ras-MAPK signaling. Moreover, gene set enrichment analysis (GSEA) showed that *Etv* deletion enhanced the activity of the MAP kinase pathway (*Figure 3H*), which is known to be decreased in FGF receptor mutants. Therefore, inactivation of FGF receptors or Etv transcription factors in the lens resulted in diametrically different outcomes in MAPK signaling.

### *Etv* deficiency blocks aberrant but not normal lens differentiation

We took three approaches to investigate the role of *Etv* transcription factors in lens cell differentiation: (1) examining the *Etv* loss-of-function phenotype, (2) corroborating our results with ERK signaling knockouts and (3) performing genetic epistasis using a FGF gain-of-function model. It was previously reported that FGF signaling controls the expression of $\alpha$A-crystallin (*Cryaa*) in the lens cells via *Etv5* and *Maf*; the latter also contains Etv-binding sites in its regulatory region (*Xie et al., 2016*). Unexpectedly, immunostaining showed that both Maf and $\alpha$-crystallins proteins were still present in *Etv* TKO mutant lenses (*Figure 4A,B,F and G*). Similarly, we did not detect a significant reduction in $\gamma$A-crystallin expression (*Figure 4C and H*), which is normally induced after $\alpha$A-crystallin

in more mature lens fiber cells. To avoid the possibility that protein perdurance may obscure the dynamic changes in gene expression, we also performed RNA in situ hybridization, which showed abundant expression of *Cryaa* and *Cryga* mRNAs in *Etv* TKO lenses (*Figure 4D,E,I and J*). Consistent with this, there were no statistically significant changes in the transcript levels of *Cryaa*, *Cryga* and *Maf* in our RNA sequencing data. Based on these results, lens differentiation can apparently proceed in the absence of *Etv* transcription factors.

To corroborate the lack of lens differentiation phenotype in *Etv* loss-of-function mutants, we next examined ERK signaling knockouts. We reasoned that, since the entire *Ets* superfamily transcription factors including *Etv* are under the control of ERK, inactivation of MAPK may produce a stronger phenotype than *Etv* knockouts. Indeed, combined deletion of *Erk1*(*Mapk3*) and *Erk2*(*Mapk1*) resulted in a drastic reduction in lens size as previously reported (*Xie et al., 2016*). However, Maf, α- and γ-crystallins mRNA and protein were still expressed in *Erk* CKO mutant lenses (*Figure 4K–O*). To further confirm this finding, we also examined the lens-specific knockout of *Mek1* (*Map2k1*) and *Mek2* (*Map2k2*), two kinases upstream of *Erk*. In the severely diminished *Mek* CKO mutant lenses, we again failed to detect significant reduction in the intensity of Maf, α- and γ-crystallins staining (*Figure 4P–T*). Therefore, genetic ablation of Erk and Etv transcription factors did not abolish lens differentiation as in FGF receptor mutants.

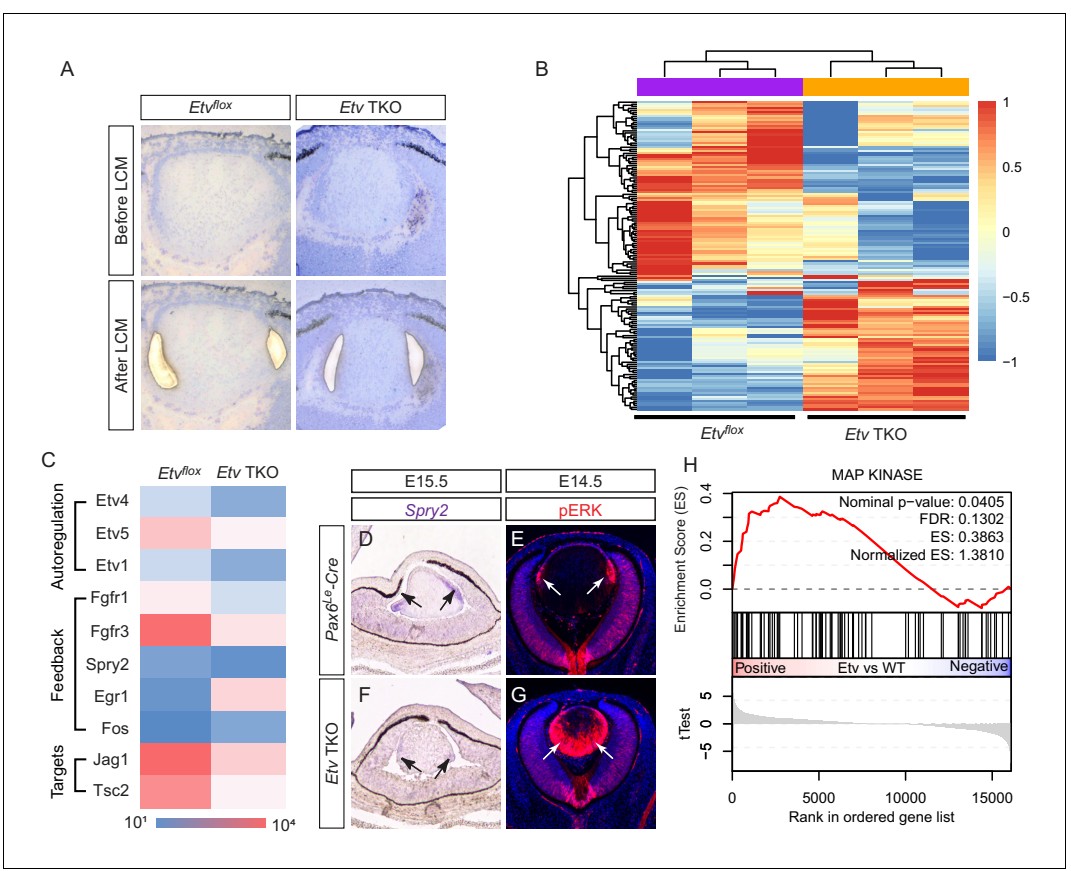

**Figure 3.** Transcriptomic analysis shows ERK signaling dysregulation in *Etv* mutant lens. (A) The transitional zone of the lens was isolated by laser capture microscope (LCM) for RNA sequencing analysis. (B) Cluster analysis of the top differentially expressed genes in the RNA sequencing data. (C) Heat map of the *Etv* regulated genes. (D and F) *Spry2* is significantly down-regulated in *Etv* TKO mutants. (H) Gene set enrichment analysis (GSEA) indicates the MAPK pathway is elevated.

The online version of this article includes the following source data and figure supplement(s) for figure 3:

**Source data 1.** Source data for *Figure 3C*.

**Figure supplement 1.** RNA sequencing analysis identified Etv-regulated genes.

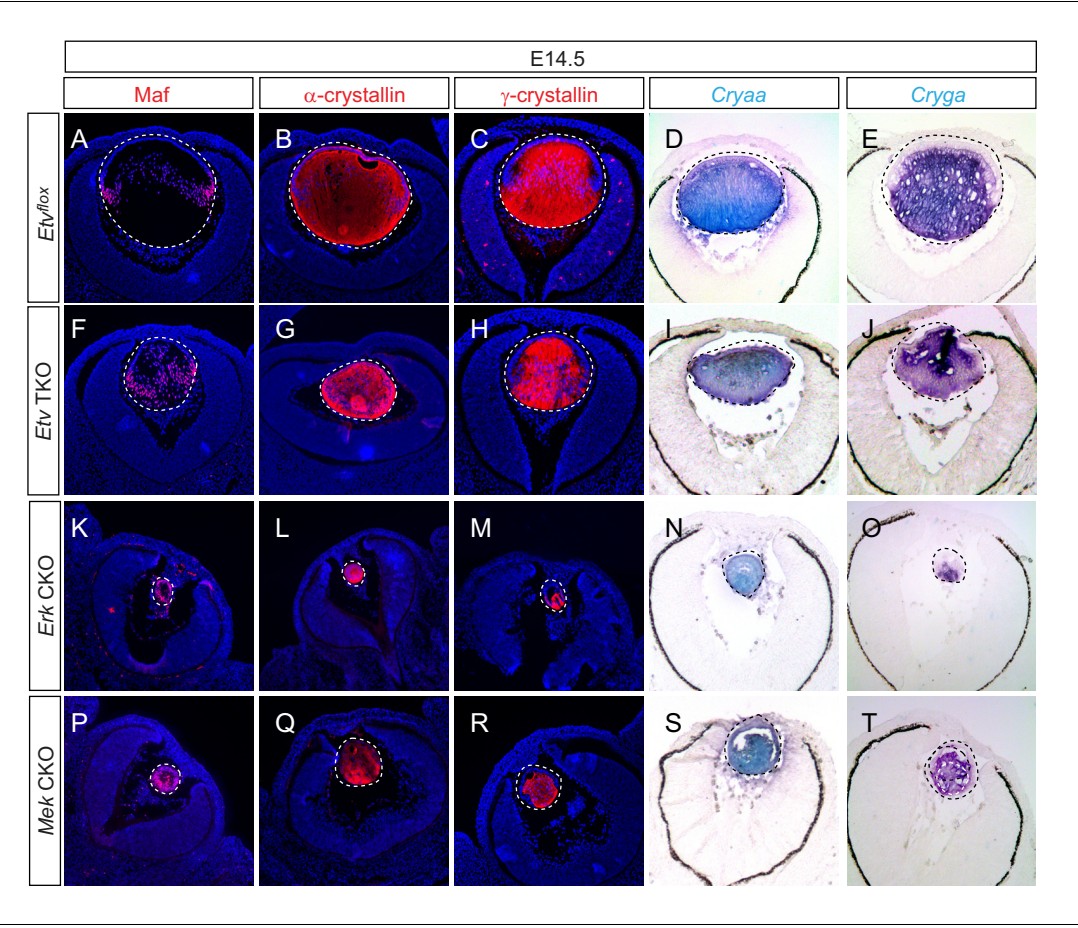

**Figure 4.** Lens fiber differentiation proceeds in the absence of *Etv* and ERK. (**A–J**) Expression of fiber cell markers Maf, α- and γ-crystallin is unaffected in *Etv* TKO lenses. (**K–T**) Maf, α- and γ-crystallin are still expressed in *Erk* and *Mek* CKO lens despite the severely reduced lens size.

The phenotypic dichotomy between *Fgfr* and *Etv* mutants raised the question whether Etv transcription factors play any role in the FGF-induced lens fiber differentiation. To address this issue, we took a gain-of-function approach to test the genetic epistasis between Etv and FGF signaling. As we and others have previously reported (*Collins et al., 2018*; *Li et al., 2019*; *Robinson et al., 1998*), the lens-specific overexpression of *Fgf3* in *Fgf3^{OVE391}* transgenic animals greatly elevated ERK phosphorylation in the lens (*Figure 5A and B*). Consistent with regulation of *Etv* and *Spry2* by ERK signaling we showed above, this led to increased expression of *Etv1*, *Etv4*, *Etv5* and *Spry2* in the *Fgf3^{OVE391}* lens (*Figure 5—figure supplement 1*), which was accompanied by ectopic induction of Maf and γ-crystallin but loss of E-cadherin in Pax6-positive anterior lens epithelium (*Figure 5F,J and N*, arrows). We hypothesized that if Etv is required for FGF signaling to promote lens differentiation, then blocking Etv activity should prevent the *Fgf3*-overexpression phenotype. Since it was cumbersome to use the triple knockout of *Etv1/4/5* to generate compound mutants for the genetic epistasis experiments, we employed the *R26^{EtvEnR}* allele, which can be induced to express an Etv4 fusion protein linked to the Engrailed repressor domain (EtvEnR) after Cre-mediated excision of a transcriptional STOP cassette. This was previously shown to act as a dominant negative protein to suppress the activities of endogenous Etv transcription factors (*Mao et al., 2009*). Consistent with this, *Pax6^{Le}-Cre;R R26^{EtvEnR}* lenses were reduced in size like those of *Etv* TKO mutants, while the expression of E-cadherin, Maf and γ-crystallin was maintained (*Figure 5G,K,O and Q*). After crossing with *Fgf3^{OVE391}*, the lens remained smaller in the *Pax6^{Le}-Cre; Fgf3^{OVE391}; R26^{EtvEnR}* mutant than that of the control (*Figure 5Q*). Of note, the expression of *Etv1*, *Etv4*, *Etv5* and *Spry2* in the *Pax6^{Le}-Cre; Fgf3^{OVE391}; R26^{EtvEnR}* mutant was diminished but not abrogated compared to that in *Fgf3^{OVE391}*

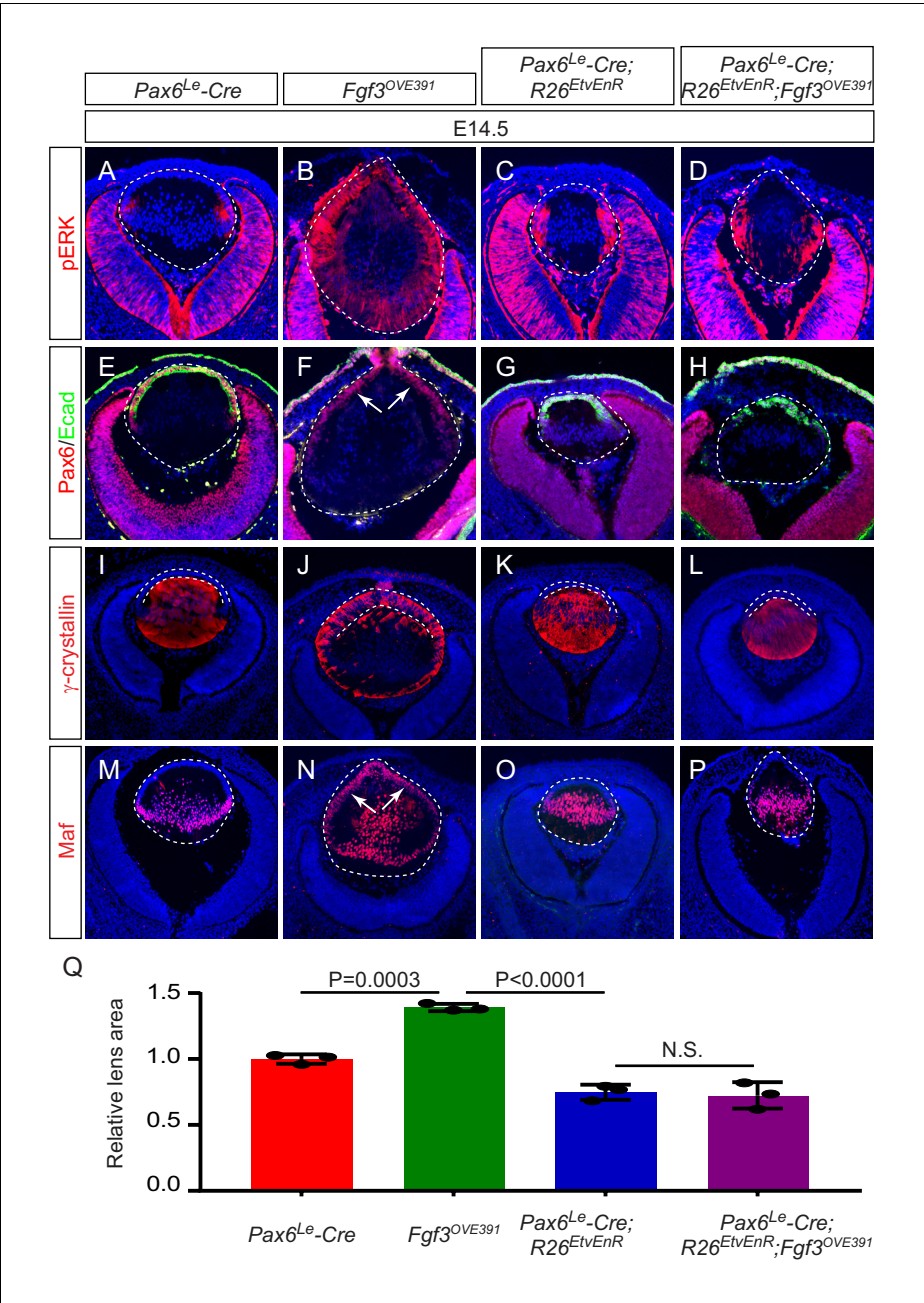

**Figure 5.** *Etv* deletion prevents FGF from inducing aberrant differentiation of the lens epithelium. (**A–P**) Overexpression of Fgf3 in *Fgf3^{OVE391}* lens induced expansion of ERK phosphorylation into the anterior lens epithelium (**B**), which lost the epithelial marker E-cadherin (**F**, arrows) but expressed the fiber cell markers Maf and γ-crystallin (**J** and **N**). The expression of dominant negative EtvEnR restored normal lens differentiation pattern in *Pax6^{Le}-Cre; Fgf3^{OVE391};R26^{EtvEnR}* lenses (**H, L and P**). (**Q**) The lens size was enlarged in *Fgf3^{OVE391}* embryos, but reduced in both *Pax6^{Le}-Cre;R26^{EtvEnR}* and *Pax6^{Le}-Cre; Fgf3^{OVE391};R26^{EtvEnR}* embryos.

The online version of this article includes the following source data and figure supplement(s) for figure 5:

**Source data 1.** Source data for *Figure 5Q*.

**Figure supplement 1.** Fgf3 overexpression stimulates expressions of *Etv* and *Spry* in the lens.

(*Figure 5—figure supplement 1*), suggesting that Etv activities was partially inhibited by *EtvEnR*. Nevertheless, in sharp contrast to *Fgf3*$^{OVE391}$, Maf and γ-crystallins were restricted to the lens fiber compartment in *Pax6*$^{Le}$-*Cre; Fgf3*$^{OVE391}$; *R26*$^{EtvEnR}$ lenses, while E-cadherin was preserved in the anterior lens epithelium (*Figure 5H,L and P*). Therefore, inhibition of Etv transcriptional activities prevented FGF signaling from inducing ectopic expression of the fiber-specific genes in the lens epithelium. Taken together, these results showed that *Etv* transcription factors were not essential for normal lens differentiation at the transitional zone, but they were required for FGF signaling to directly convert lens epithelial cells into lens fibers.

## Etv regulates the induction of cell differentiation by promoting Jag1-Notch signaling

Although the fiber cell differentiation genes were still expressed in *Etv* TKO mutant lenses, we noticed that the transitional zone marked by the boundary between the lens epithelial marker Foxe3 and the fiber cell marker Prox1 was moved anteriorly (*Figure 6A and E*, arrows). As a result, the length of the lens epithelium marked by Pax6 and E-cadherin staining was shortened compared to the circumference of the posterior lens (*Figure 6B,F and I*, arrows). This differentiation pattern is reminiscent of defective Notch signaling, which is known to cause the lens progenitor cells to undergo premature differentiation before reaching the lens equator, resulting in an anteriorly shift of the transitional zone (*Jia et al., 2007*; *Le et al., 2009*; *Li et al., 2019*; *Rowan et al., 2008*; *Saravanamuthu et al., 2012*). Intriguingly, one of the down-regulated genes in *Etv* TKO mutants revealed by our transcriptomic analysis was *Jag1* (*Figure 3C*), which encodes the ligand for Notch signaling. By immunostaining, we confirmed that the expression of Jag1 protein was drastically reduced in *Etv* TKO mutant lenses (*Figure 6C and G*, arrowheads). Jag1 expressed in the lens fiber cells signals to the Notch receptors in the lens epithelium to maintain the progenitor pool (*Jia et al., 2007*; *Le et al., 2009*). In line with the reduced Notch signaling due to Jag1 deficiency, the anterior lens epithelium in *Etv* TKO mutants exhibited severely diminished staining of the Notch1 intracellular domains (Notch1-ICD), a proteolytic product of Notch1 receptor triggered by Jag1 activation (*Figure 6D and H*, arrows). These results support that Etv regulates Jag1-Notch signaling to control the timing of lens cell differentiation.

To investigate the molecular mechanism of Jag1 regulation by Etv, we turned to cultured lens cells in vitro. By western blot, we showed that Jag1 expression in the lens epithelial cell culture was induced by FGF2 after 5 hr treatment and suppressed by Mek inhibitors U0126 and PD0325901 but not PI3K inhibitor LY294002 (*Figure 6J* and *Figure 6—figure supplement 1*). This is consistent with previous reports that Jag1 expression in the lens is controlled by ERK-mediated FGF signaling (*Li et al., 2019*; *Saravanamuthu et al., 2009*). To determine whether Jag1 is a direct transcriptional target of Etv, we used the recently published ATAC-seq analysis (*Zhao et al., 2019*) to search the open chromatin regions in the developing lens and identified two Etv-binding sites located within introns 2 and 5 of *Jag1* gene (*Figure 6K*). By chromatin immunoprecipitation (ChIP), we showed that both introns 2 and 5 sites could be pulled down by Etv5 antibodies but not by IgG control (*Figure 6L*), demonstrating that these two sites in *Jag1* were occupied by Etv in the lens cells. We next investigated whether Etv was required for FGF signaling to induce Jag1 expression in vivo. Jag1 was normally restricted to the nascent lens fibers in wild type control lenses (*Figure 5M*, arrowheads), but it was abolished in *Mek* and *Erk* CKO mutants (*Figure 6N and O*). In *Fgf3*$^{OVE391}$ lenses, however, Jag1 expression was expanded to the entire lens epithelium at the expense of E-cadherin expression (*Figure 6P*, arrow), demonstrating that FGF signaling can induce de novo Jag1 expression in the lens epithelium. In *Pax6*$^{Le}$-*Cre; R26*$^{EtvEnR}$ lenses, Jag1 expression was significantly reduced compared to those of wild-type control (*Figure 6Q*, arrowheads), consistent with the role of Etv in Jag1 regulation. Suppression of Etv activity in *Pax6*$^{Le}$-*Cre; Fgf3*$^{OVE391}$; *R26*$^{EtvEnR}$ lenses reversed the Fgf3 overexpression phenotype, limiting the Jag1 expression domain to the posterior lenses (*Figure 6R*, arrowheads). Based on these results, we conclude that Etv mediates the FGF-ERK-Notch crosstalk to control the induction of lens differentiation.

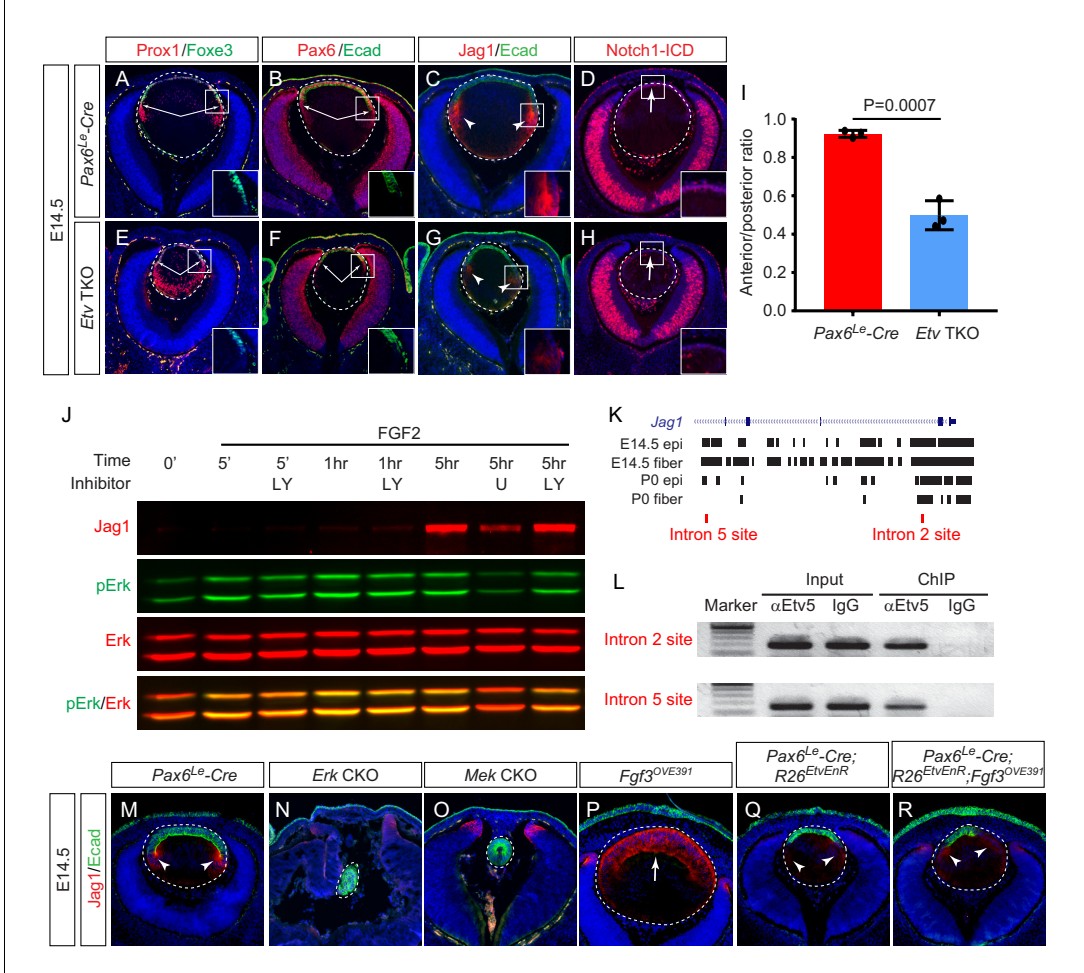

**Figure 6.** Etv induces Jag1 expression to control Notch signaling. (**A–H**) The transitional zone marked by the boundaries of Prox1, Foxe3, Pax6 and E-cadherin expressions are shifted anteriorly in *Etv* TKO mutant lenses (A, B, E and F, arrows). This was caused by reduced expression of Jag1 in the nascent lens fibers (C and G, arrowheads) and down regulation of Notch signaling as indicated by Notch1-ICD staining in the lens epithelium (D and H, arrows). (**I**) Quantification of the anterior and posterior perimeters of the lens shows the relative shortening of the lens epithelium. (**J**) FGF2 induced Jag1 expression in the lens culture after 5 hr, which was blocked by Mek inhibitor U1206, but not by PI3K inhibitor LY294002. (**K**) Bioinformatic analysis identifies two Etv binding sites within the intron 2 and 5 of the *Jag1* locus. The bars indicate the open chromatin regions obtained from the ATAC-seq analysis of lens epithelium and fibers. (**L**) Chromatin immunoprecipitation experiment in lens cultures showed that both the introns 2 and 5 sites were pulled down by Etv5 antibody but not IgG control. (**M–R**) The endogenous Jag1 expression in lens fiber cells (M, arrowheads) is abolished by deletion of either *Erk* or *Mek* (N and O). Overexpression of Fgf3 induces ectopic Jag1 expression in the lens epithelium (P, arrow), which is suppressed by dominant negative EtvEnR (Q and R, arrowheads).

The online version of this article includes the following source data and figure supplement(s) for figure 6:

**Source data 1.** Source data for *Figure 6I*.

**Figure supplement 1.** FGF induces Jag1-Notch pathway in an ERK-dependent manner.

## Deletion of *Etv* genes led to aberrant activation of mTOR signaling and disruption of nuclei clearance in the lens

The final step of lens cell differentiation is the degradation of their cellular organelles, which is crucial for the transparency of the lens (*Bassnett et al., 2011*). To determine whether *Etv* transcription factors are required for this process, we examined the clearance of lens cell nuclei by histology. In wild-type control, the maturing fiber cells lost their nuclei as they migrated toward the interior of the lens, eventually forming an organelle free zone (OFZ) (*Figure 7A* and *Figure 7—figure supplement 1*, arrowheads). In contrast, *Etv* TKO mutants retained nuclei within the lens core, indicating defective organelle clearance (*Figure 7E* and *Figure 7—figure supplement 1*, arrowheads). Previous

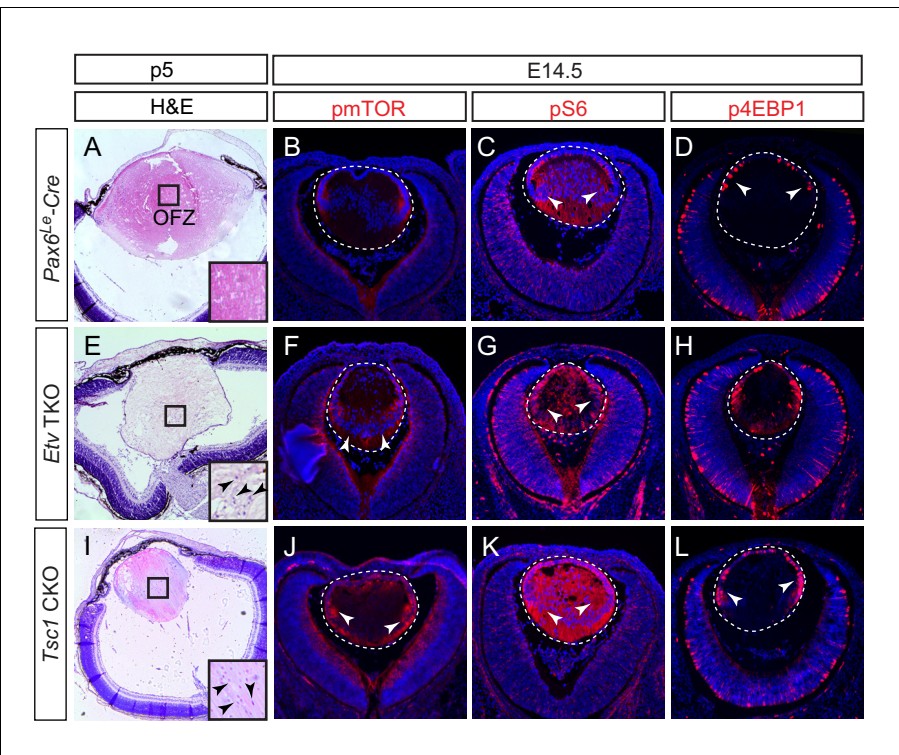

**Figure 7.** Activation of mTOR signaling disrupts nuclei degradation in the *Etv* mutant. (**A–J**) The center of the wild type lens was an organelle free zone (OFZ) at P5 (**F**), but it still contains nuclei in *Etv* mutants (F, arrowheads). This is associated with increased mTOR signaling as indicated by the elevated phosphorylation of mTOR, S6 and 4EBP1 (B-D, F-H, arrowheads). (**I–L**) The aberrant retention of nuclei and activation of mTOR signaling are reproduced in *Tsc1* knockout lens.

The online version of this article includes the following source data and figure supplement(s) for figure 7:

**Figure supplement 1.** Persistent nuclei in *Etv* and *Tsc1* mutant lenses.

**Figure supplement 1—source data 1.** Source data for *Figure 7—figure supplement 1D*.

studies have shown that inhibition of mTOR signaling accelerated organelle elimination in chick lens explants (*Basu et al., 2014*). Interestingly, our RNA sequencing analysis of *Etv* TKO mutant lenses showed significant down-regulation of Tsc2, which forms a heterodimer with Tsc1 to inhibit mTOR activity (*Figure 3C*). This suggests that the defective nuclei clearance in the *Etv* TKO mutant may be caused by elevated mTOR signaling. In support of this hypothesis, we observed that phospho-mTOR (pmTOR) was localized in the transitional zone of the wild-type control lens, but it expanded into the fiber cell compartment in *Etv TKO* mutants (*Figure 7B and F*, arrowheads). Similarly, *Etv* TKO mutant lenses displayed significant increase in pS6 and p4EBP1, two known substrates of mTOR kinase (*Figure 7C,D,G and H*). Therefore, inactivation of *Etv* genes led to increased mTOR signaling in the lens.

To determine whether mTOR activation indeed interfered with nuclei clearance in the lens, we generated *Pax6^Le-Cre; Tsc1^flox/flox* (*Tsc1* CKO) to ablate the Tsc2 binding partner Tsc1. As expected from the role of Tsc1/2 complex in suppressing mTOR activity, pmTOR was significantly increased in *Tsc1* CKO lenses, which also displayed elevated level of pS6 and p4EBP1 (*Figure 7J–L*). Importantly, whereas the wild-type control lens contains a well-defined OFZ in the center, the nuclei were scattered throughout the *Tsc1* CKO lenses (*Figure 7I* and *Figure 7—figure supplement 1*). The lack of an OFZ in *Tsc1* deficient lenses supports our model that Etv control of mTOR signaling is required for the terminal differentiation of the lens fiber cells.

## Discussion

FGF signaling plays important roles during vertebrate lens development. As demonstrated by the profound lens defects in FGF receptor mutants, the primary function of FGF signaling is to promote differentiation of the lens epithelial cells into the fiber cells. In this study, we have studied the functions of *Etv* family genes, which are the well-established downstream transcription factors induced by FGF during embryonic development. Contrary to the defective lens differentiation found in FGF signaling mutants, deletion of *Etv* genes did not prevent the expression of lens fiber genes in normal development. Instead, it caused premature differentiation of the lens progenitors as a result of reduced Notch signaling. Moreover, *Etv* mutations led to an increase in the activity of ERK and mTOR signaling, with the latter responsible for the abnormal retention of the nuclei in the adult lens. Therefore, unlike BMP, Hedgehog and Wnt signaling that command their downstream signal-regulated transcription factors (SRTFs) as faithful effectors, ETV transcription factors activated by FGF signaling can oppose the intended function of the pathway.

Our study of *Etv* mutants in lens development has revealed several important features of *Etv* function that deviate from the expected role of a faithful executor for FGF signaling. First, deletion of *Etv* caused significant derangement of ERK signaling, which is the main intracellular pathway activated by FGF. Contrary to the loss of ERK activity in mutants lacking either FGF receptors or downstream mediators Frs2 and Shp2 (*Collins et al., 2018*; *Li et al., 2014*; *Pan et al., 2010*; *Zhao et al., 2008*), *Etv*-deficient lenses displayed elevated ERK signaling, which was evident in the increased level of ERK phosphorylation and expression of ERK early response genes *Egr1* and *Fos*. This was likely due to reduced expression of ERK inhibitor *Spry2*, which has been shown to be a direct transcriptional target of Etv (*Yan et al., 2013*). In support of this, deletions of *Spry* genes have been found to result in upregulation of ERK signaling in a wide variety of tissues, including the lens (*Kuracha et al., 2011*). The aberrant ERK activation in *Etv* mutants may stimulate downstream targets normally silenced by loss of FGF signaling, inducing further compensatory changes that rewire the signaling network. It underscores the intricate role of Etv in stabilizing the intracellular signaling activated by FGF.

The second distinction of *Etv* mutants compared to FGF receptor knockouts is the relatively normal lens differentiation. This was unexpected because a previous study shown that both lens differentiation genes *Maf* and *Cryaa* were under direct control of Etv transcription factors in vitro (*Xie et al., 2016*). We presented three lines of evidence in support of our conclusion. First, we performed RNA sequencing, RNA in situ hybridization, and immunohistochemistry in *Etv* mutant lenses, none of which showed significant reduction in the expression of *Maf* and *Cryaa*. Second, we generated lens-specific knockouts of *Erk1/2*, which we showed to be required for *Etv* expression. Even in these ERK signaling mutants, we still observed expression of *Maf* and *Cryaa*. Of note, the previous study observed that immunostaining for α-crystallin and Maf were severely reduced or even absent in the *Erk1/2* knockout lens (*Xie et al., 2016*). It is not clear what caused the phenotypic discrepancy between ours and the previous study, which could be due to a difference in the genetic background of the animal models or the source of antibodies. It should be noted that γ-crystallin expression appeared to move posteriorly in both *Etv* and *Erk* knockouts (*Figure 4H,M and R*), which may reflect a slight delay of lens fiber terminal differentiation. Nevertheless, even knockout of ERK kinase *Mek* still failed to abolish *Maf* and *Cryaa* expression. In the third approach, we explored the role of Etv in a gain-of-function setting, showing that inhibition of *Etv* indeed blocked ectopic FGF from stimulating Maf and γ-crystallin expression in the lens epithelium. This result suggests an intrinsic difference between normal differentiation at the lens equator and ectopic differentiation in the lens epithelium, which may be shaped by their unique microenvironment and different dosage of FGF signaling. Taken together, our results demonstrate that *Etv* genes are dispensable for FGF-induced cell differentiation under the physiological condition during lens development.

Genetic ablation of *Etv* also differed from inactivation of FGF signaling in its impact on the transitional zone of cell differentiation, which was shifted anteriorly after *Etv* deletion and posteriorly in FGF signaling mutants. We recently showed that FGF plays a dual role in specifying the location of the transitional zone by regulating the timing of lens differentiation; FGF directly promotes differentiation of lens fiber cells, but it also cooperates with PDGF to restrain differentiation of the lens progenitor cells (*Li et al., 2019*). The latter is mediated by Jag1, which induces Notch signaling in the anterior lens epithelium to suppress progenitor cell differentiation. In the current study, we

determined that Jag1 expression is under the direct control of Etv transcription factors and ERK, delineating a complete molecular cascade that connects FGF to Notch signaling. In the absence of *Etv*, loss of Notch signaling leads to premature differentiation of lens progenitor cells indicated by Foxe3 and Prox1 expression, but the pattern of Crystallin expressions are unchanged. This suggests the terminal differentiation of lens fibers may be slightly delayed relative to onset of progenitor cells differentiation. Nevertheless, *Etv* mutants proceed to express *Crystallin* genes at the comparable level as wild-type controls, which is in sharp contrast to the loss of γ-crystallin expression in FGF receptor mutants (*Zhao et al., 2008*). Therefore, Etv mediates the anti-differentiation function of FGF to activate Notch signaling but it is largely dispensable for the pro-differentiation aspect of FGF signaling.

Lastly, our study showed that *Etv* deficiency augmented mTOR phosphorylation. This is likely caused by reduced expression of *Tsc2*, a negative regulator of mTOR. Indeed, we also observed significant increase in phosphorylation of S6 and 4EBP1, two downstream targets of mTOR. Importantly, we observed aberrant retention of the lens fiber cell nuclei in postnatal *Etv* mutants, which was mimicked by deletion of Tsc2 partner Tsc1 in the lens. Previous studies have shown that inhibition of mTOR by rapamycin promotes degradation of the organelle in chick lens explants although the exact mechanism remained controversial (*Basu et al., 2014*; *Morishita and Mizushima, 2016*). Our study provides the in vivo evidence that mTOR is indeed an important regulator of organelle removal in lens maturation.

Taken together, our genetic study firmly establishes *Etv* family transcription factors as SRTFs for FGF signaling in lens development. SRTFs constitute a special class of transcription factors that relay intracellular signals to shape the cellular transcriptome, which ultimately determines the identity and status of the cell. Although many SRTFs have proven to be faithful to their upstream signaling, we showed that *Etv* transcription factors deviate or even oppose the function of FGF-ERK signaling in the lens. This is likely because, unlike other signaling pathways such as BMP, Hedgehog and Wnt that rely on a single downstream effector, FGF signaling can induce multiple transcription factors. This generates diversity among the FGF SRTFs that may evolve to acquire their unique or even opposing functions, which fine tunes the overall transcriptional response. The three *PEA3/ETV* genes and *ERG* are the only *ETS* genes implicated in tumorigenesis, which has been attributed to their ability to mimic the oncogenic effect of Ras signaling (*Hollenhorst et al., 2011*). Our previous work in lacrimal gland development has indeed found that deletion of *Etv* transcription factors largely reproduced the phenotype of FGF-ERK mutants (*Garg et al., 2018*). In the lens, however, our current study paints a more complex picture of Etv activity, suggesting that their functions deviate considerably from their upstream FGF signaling. As transcription factors attract increasing interests as viable therapeutic targets, it will be important to elucidate the context-dependent function of these SRTFs in development and diseases.

## Materials and methods

**Key resources table**

| Reagent type (species) or resource | Designation | Source or reference | Identifiers | Additional information |
|---|---|---|---|---|
| Antibody | Rabbit monoclonal anti-cyclin D1 | Cell Signaling | Cat.# 55506, RRID:AB_2827374 | IHC-1/200 |
| Antibody | Rabbit polyclonal anti-Maf | Santa Cruz Biotechnology | Cat.#sc-7866, RRID:AB_638562 | IHC-1/200 |
| Antibody | Mouse monoclonal anti-Ecadherin | BD | Cat.#610181, RRID:AB_397580 | IHC-1/500 |
| Antibody | Rabbit monoclonal anti-Erk | Cell Signaling | Cat.# 4695, RRID:AB_390779 | WB-1/2000 |
| Antibody | Mouse monoclonal anti-Foxe3 | Santa Cruz Biotechnology | Cat.#sc-377465 | IHC-1/200 |
| Antibody | Rabbit polyclonal anti-Jag1 | Santa Cruz Biotechnology | Cat.#sc-8303, RRID:AB_649685 | IHC-1/200, WB-1/500 |

*Continued on next page*

*Continued*

| Reagent type (species) or resource | Designation | Source or reference | Identifiers | Additional information |
|---|---|---|---|---|
| Antibody | Mouse monoclonal anti-Ki67 | BD | Cat.#550609, RRID:AB_393778 | IHC-1/200 |
| Antibody | Rabbit monoclonal Lamin A/C | Abcam | Cat.# ab133256, RRID:AB_2813767 | IHC-1/1000 |
| Antibody | Rabbit monoclonal Notch1 | Cell Signalling | Cat.#4380, RRID:AB_10691684 | WB-1/1000 |
| Antibody | Rabbit monoclonal Notch1-ICD | Cell Signaling | Cat.#4147, RRID:AB_2153348 | IHC-1/200 |
| Antibody | Rabbit polyclonal anti-Prox1 | Covance | Cat.#PRB-238C, RRID:AB_291595 | IHC-1/200 |
| Antibody | Rabbit polyclonal anti-pHH3 | Millipore | Cat.#06-570, RRID:AB_310177 | IHC-1/200 |
| Antibody | Rabbit anti-pS6 | Cell Signaling | Cat.#5364, RRID:AB_10694233 | IHC-1/200 |
| Antibody | Rabbit monoclonal anti-pmTOR | Cell Signaling | Cat.#5536, RRID:AB_10691552 | IHC-1/200 |
| Antibody | Rabbit polyclonal anti-Pax6 | Covance | Cat.#PRB-278P, RRID:AB_291612 | IHC-1/200 |
| Antibody | Rabbit anti-p4EBP1 | Cell Signaling | Cat.#2855, RRID:AB_560835 | IHC-1/200 |
| Antibody | Rabbit monoclonal anti-pERK | Cell Signaling | Cat.#9101, RRID:AB_331646 | IHC-1/200 |
| Antibody | Mouse monoclonal anti-pERK | Santa Cruz Biotechnology | Cat# sc-7383, RRID:AB_627545 | WB-1/1000 |
| Antibody | Rabbit polyclonal anti-α-Crystallin | Sam Zigler (National Eye Institute) | | IHC-1/5000 |
| Antibody | Rabbit polyclonal anti-γ-Crystallin | Sam Zigler (National Eye Institute) | | IHC-1/5000 |
| EdU | DNA synthesis monitoring probe | Abcam | cat. # ab14618 | |
| Commercial assay Kit | Click IT EdU Cell proliferation kit | Invitrogen | cat. # C10337 | |
| Commercial assay Kit | In situ cell death detection kit | Roche | cat.# 1168479510 | |
| Peptide, recombinant protein | recombinant murine FGF2 | ScienCell | cat.# 124-02 | |
| Genetic reagent (*M. musculus*) | *Etv1*<sup>flox</sup> | PMID:12741988 | MGI:2663693 | Dr. Silvia Arber (University of Basel, Basel, Switzerland) |
| Genetic reagent (*M. musculus*) | *Etv4*<sup>-/-</sup> | PMID:11094084 | MGI:2445834 | Dr. Xin Sun (University of California at San Diego, San Diego, CA) |
| Genetic reagent (*M. musculus*) | *Etv5*<sup>flox</sup> | PMID:19386269 | MGI:3849047 | Dr. Xin Sun (University of California at San Diego, San Diego, CA) |
| Genetic reagent (*M. musculus*) | *Tsc1*<sup>flox</sup> | PMID:12205640 | MGI:2656240 | Dr. Stephen Tsang (Columbia University, New York, NY) |

*Continued on next page*

Continued

| Reagent type (species) or resource | Designation | Source or reference | Identifiers | Additional information |
|---|---|---|---|---|
| Genetic reagent (*M. musculus*) | *R26*[EtvEnR] | PMID:19386268 | MGI:3848910 | Drs. Andrew McMahon (University of Southern California, Los Angeles, CA) and James Li (University of Connecticut Health Center, Farmington, CT) |
| Genetic reagent (*M. musculus*) | *Map2k1*[flox] | PMID:16887817 | MGI:3714918 | |
| Genetic reagent (*M. musculus*) | *Map2k2*[KO] | PMID:12832465 | MGI:2668345 | |
| Genetic reagent (*M. musculus*) | *Mapk3*[-/-] | PMID:11160759 | MGI:3042006 | |
| Genetic reagent (*M. musculus*) | *Mapk1*[flox] | PMID:18596172 | MGI:3803954 | |
| Genetic reagent (*M. musculus*) | *Fgf3*[OVE391] | PMID:7539358 | MGI:6393977 | Dr. Michael Robinson, Miami University. |
| Genetic reagent (*M. musculus*) | *Pax6*[Le]-*Cre* | PMID:11069887 | MGI:3045795 | |
| Genetic reagent (*M. musculus*) | *Fgfr1*[flox] | PMID:16421190 | Stock #: 007671 MGI:3713779 | |
| Genetic reagent (*M. musculus*) | *Fgfr2*[flox] | PMID:12756187 | MGI:3044679 | Dr. David Ornitz, Washington University Medical School, St Louis, MO |

## Mice

Mice carrying *Erk1(Mapk3)*[-/-], *Mapk1*[flox], *Mek1(Map2k1)*[flox] and *Mek2(Map2k2)*[-/-] alleles were bred and genotyped as described (*Newbern et al., 2008*; *Newbern et al., 2011*). *Pax6*[Le]-*Cre* mice were from Dr. Ruth Ashery-Padan (Tel Aviv University, Tel Aviv, Israel), *Etv1*[flox] mice from Dr. Silvia Arber (University of Basel, Basel, Switzerland), *Etv4*[+/-] and *Etv5*[flox] mice from Dr. Xin Sun (University of California at San Diego, San Diego, CA), *R26*[EtvEnR] from Drs. Andrew McMahon (University of Southern California, Los Angeles, CA) and James Li (University of Connecticut Health Center, Farmington, CT), *Fgf3*[OVE391] mice were obtained from Dr. Michael Robinson (Miami University, Oxford, OH) and *Fgfr2*[flox] from Dr. David Ornitz (Washington University Medical School, St Louis, MO) (*Ashery-Padan et al., 2000*; *Mao et al., 2009*; *Patel et al., 2003*; *Robinson et al., 1998*; *Yu et al., 2003*; *Zhang et al., 2009*). *Fgfr1*[flox] mice were from Jackson Laboratory (Stock No: 007671). *Tsc1*[flox] mice were originally obtained from Jackson Laboratory (Stock No:005680) and provided by Dr. Stephen Tsang (Columbia University, New York, NY). To generate embryos, females in breeding were checked for vaginal plugs (considered as 0.5 days pc). All the animals were maintained in the mixed genetic background. Mouse maintenance and experimentation was performed according to protocols approved by Columbia University Institutional Animal Care and Use Committee. *Pax6*[Le]-*Cre* or *Etv*[flox] mice did not display any lens phenotypes and were used as controls. All the experiments were repeated at least three times.

## Histology

Hematoxylin and Eosin staining (H and E) was performed as previously described (*Carbe et al., 2012*). Briefly, paraffin blocks were sections at 10 μm and deparaffinized by heating and histosol washes, followed by rehydration through decreasing percentage of ethanol solutions. The slides were dipped into hematoxylin for 3 min followed by 10-15 min wash with tap water. The samples were decolorized with 1% acid alcohol for 15 s, before treatment with Eosin for 1 min. Samples were

then dehydrated by passing through increasing concentration of ethanol, and transferred to histosol. The samples were mounted using permount mounting medium.

## Immunohistochemistry

For immunohistochemistry, paraffin samples were deparaffinized as described above and cryosections were briefly washed with PBS to remove OCT. Antigen retrieval was performed with microwave boiling for 1-2 min followed by heating for 10 min at low-power settings in citrate buffer (10 mM sodium citrate, pH 6.0). Sections were then washed with PBS and blocked with 5% NGS/0.1% Triton in PBS. Primary antibody incubation was performed overnight at 4°C in humid chamber followed by incubation with florescent-conjugated secondary antibodies for 1 hr at room temperature in dark. The following primary antibodies were used: Pax6 (PRB-278P), Prox1 (PRB-238C) (both from Covance, Berkeley, CA), E-cadherin (U3254, Sigma, St Louis, Missouri), Maf (sc-7866), Foxe3 (sc-377465), Jag1 (sc-8303) (all from Santa Cruz Biotechnology), Lamin A/C (ab133256, Abcam), pHH3 (06-570, Millpore), phospho-4EBP1 (#2855). phospho-S6 (#5364), phospho-Akt (D9E), phospho-Erk (#4370), phospho-mTOR (#5536), Cyclin D1 (#55506), Notch1-ICD (# 4147) (all from Cell signaling Technology). Antibodies against α- and γ-crystallins were kindly provided by Dr. Sam Zigler (National Eye Institute). For Notch1-ICD antibody staining, samples were paraffin embedded, sectioned and followed by antigen retrieval for 20 min in a pressure cooker. To detect phospho-Erk, phospho-mTOR, phospho-S6, phospho-4EBP1 and NICD, HRP-conjugated secondary antibody and Tyramide signal amplification kit (Perkin Elemer) were used.

## Quantification of lens phenotype

The perimeters of the anterior epithelial layer and posterior fiber cells were measured using ImageJ and the ratio was calculated for control and mutant lenses. To quantify the nuclei degradation, the lenses were stained with nuclear markers DAPI and Lamin A/C. The number of nuclei within a concentric circle of the lens at the half of the lens radius were counted. For statistical analysis, at least three embryos of each genotype were taken and two lens sections at the equatorial plane per embryo were analyzed. The statistical significance was calculated by one-way ANOVA.

## TUNEL assay

TUNEL assays were performed on 10-µm paraffin sections following the manufacturer's instructions in the Fluoroscein In Situ Cell Death Detection kit (Roche Applied Science, Indianapolis, IN). Apoptosis rates were calculated as the ratio of TUNEL-positive cells to DAPI-positive cells in control and mutant samples, and results were analyzed by t-test.

## EdU incorporation assay

Pregnant females were injected with EdU (ab146186, Abcam) dissolved in DMSO at the dosage of 50 mg/kg body weight. After 2 hr, the mice were sacrificed and embryos collected for cryosection. For EdU detection, the Click-IT EdU Imaging Kit (C10337, Invitrogen) was used according to the manufacturer's instructions. The proliferation rates in the lens epithelium were calculated as the ratios of EdU-positive cells to DAPI-positive cells within the epithelium and the statistical significances between controls and mutants were evaluated by t-test.

## RNA in situ hybridization

Section in situ hybridization was performed as described (*Carbe et al., 2013*). Briefly, the cryoblocks were sectioned at 10 µm and hybridized with diluted probes (1:500) at 65°C overnight. The sections were washed 3X with wash buffer at 65°C for 30 min, followed by 2X wash with MABT for 30 min. The sections were then blocked with blocking buffer for 1 hr at room temperature, followed by incubation with anti-DIG antibody (1:1500) overnight at 4°C. Next, the slides were washed 4X with MABT for 20 min and 2X alkaline phosphatase buffer for 10 min, before incubating with BM purple for colorimetric reaction for 24 hrs at room temperature. The following probes were used: *Pea3*, *Erm5* (from Dr. Bridget Hogan, Duke University Medical Center, Durham, NC, USA), *Er81* (from Dr. Gord Fishell, New York University Medical Center, New York, NY, USA), *Fgfr1* and *Fgfr3* (from Suzanne Mansour, University of Utah, Salt Lake city, UT) and *Sprouty2* (from Gail Martin, University of California at San Francisco, San Francisco, CA).

### Laser capture micro-dissection and gene expression profiling

Freshly harvested embryos were frozen in OCT medium (Sakura Finetek), sectioned at 10 µm thickness and transferred to PEN slides (Ziess). To fix and stain the slides, they were dipped in 95% ethanol for 2 min to fix the samples, stained with crystal violet stain (3% in ethanol) on ice. The slides were then rigorously washed in 2 X 70% ethanol for 30-40 s to remove the OCT and dehydrated in 100% ethanol for 2 min. For control and *Etv* mutant embryos, the lens tissue was micro-dissected from the transitional zone using Laser capture microscope (Zeiss AxioObserver.Z1 inverted microscope). The RNA was extracted using Qiagen Micro Plus kit. Conversion to cDNA and amplification were performed using Clontech SMART-seq v4 Ultra low input RNA kit and the cDNA library construction was performed using Nextera XT DNA library preparation kit by the core facility at Columbia university prior to RNA sequencing. The RNAseq data are available from the GEO repository (GSE137215). Normalization and differential gene expression analysis for RNA-seq data were performed using De-Seq package in R studio. The GSEA analysis was performed using JavaGSEA desktop software from Broad Institute.

### Cell culture and western blot

Immortalized lens cells were authenticated by immunostaining with lens markers and confirmed to be free of mycoplasma contamination as previously described (*Li et al., 2019*). They were cultured in Dulbecco's modified Eagle's medium (DMEM) supplemented with 10% fetal bovine serum (FBS) and L-glutamine, 1% penicillin-streptomycin. After serum starved for 24 hr, cells were subject to FGF2 (50 ng/ml) stimulation, with or without Mek inhibitor U0126 (50 µM), PD0325901 (50 µM) or PI3K inhibitor LY (50 µM) for the indicated time periods, and harvested in CelLytic-M lysis buffer (Sigma) with proteinase inhibitor cocktail (Thermo fisher). Protein lysates were collected following centrifugation at 12,000 g for 10 min and resuspended in SDS buffer. Equal amounts of total protein were loaded for western blot analysis and visualized using an Odyssey SA scanner (LICOR Biosciences, Lincoln, NE). Antibodies used were Jagged1 (H-114, Santa Cruz Biotechnology), Notch1 (#4380, Cell signaling Technology), phospho-ERK1/2 (sc-7383, Santa Cruz Biotechnology), ERK (#4695, Cell signaling Technology).

### Chromatin immunoprecipitation

The Chromatin Immunoprecipitation (ChIP) assays were performed in immortalized lens cells as previously described (*Garg et al., 2017*; *Li et al., 2019*). The antibodies used were IgG as isotype control (#2729, Cell Signaling Technology) and anti-Etv5 (#66657, Proteintech). The primers used for the intron 2 site are GGTTTCTGCTCCACCTCTGA and GGGAGTGCAAACTTGATGCT, and for the intron 5 site are AAGAGCCAGCTCAGCTTCAC and AGATCTGTGCCCCAGAGGAT.

## Acknowledgements

The authors thank Drs. Silvia Arber, Ruth Ashery-Padan, James Li, Andrew McMahon, Michael Robinson, David Ornitz, Xin Sun and Stephen Tsang for mice, Drs. Bridget Hogan, Gord Fishell, Suzanne Mansour and Gail Martin for in situ probes. We also thank Drs. Carlo Maurer and Kenneth Olive for help with Laser Capture Microscopy, Drs. Howard Worman and Ji-Yeon Shin for advice on Lamin A/C antibody, Dr. Mukesh Bansal for bioinformatics analysis, Joseph Ryo for critical reading of the manuscript. The work was supported by NIH (EY017061 and EY025933 to XZ). The Columbia Ophthalmology Core Facility is supported by NIH Core grant 5P30EY019007 and unrestricted funds from Research to Prevent Blindness (RPB). XZ is supported by Jules and Doris Stein Research to Prevent Blindness Professorship. AG was a recipient of STARR fellowship. QW is supported by a Postdoctoral Fellowship from Natural Sciences and Engineering Research Council of Canada.

## Additional information

### Funding

| Funder | Grant reference number | Author |
|---|---|---|
| National Eye Institute | EY017061 | Xin Zhang |

| | | |
|---|---|---|
| National Eye Institute | EY025933 | Xin Zhang |
| Research to Prevent Blindness | Jules and Doris Stein Research to Prevent Blindness Professorship | Xin Zhang |
| Starr Foundation | Graduate fellowship | Ankur Garg |
| Natural Sciences and Engineering Research Council of Canada | Postdoctoral fellowship | Qian Wang |

The funders had no role in study design, data collection and interpretation, or the decision to submit the work for publication.

### Author contributions

Ankur Garg, Data curation, Formal analysis, Investigation, Methodology; Abdul Hannan, Qian Wang, Neoklis Makrides, Hongge Li, Investigation; Jian Zhong, Resources; Sungtae Yoon, Data curation, Formal analysis, Investigation; Yingyu Mao, Investigation, Writing - review and editing; Xin Zhang, Conceptualization, Resources, Formal analysis, Supervision, Funding acquisition

### Author ORCIDs

Xin Zhang  https://orcid.org/0000-0001-5555-0825

### Ethics

Animal experimentation: This study was performed in strict accordance with the recommendations in the Guide for the Care and Use of Laboratory Animals of the National Institutes of Health. All of the animals were handled according to approved institutional animal care and use committee (IACUC) protocol (AABD8562) of Columbia University Medical Center.

### Decision letter and Author response

Decision letter https://doi.org/10.7554/eLife.51915.sa1
Author response https://doi.org/10.7554/eLife.51915.sa2

## Additional files

### Supplementary files

• Transparent reporting form

### Data availability

The RNAseq data are available from the GEO repository (GSE137215).

The following dataset was generated:

| Author(s) | Year | Dataset title | Dataset URL | Database and Identifier |
|---|---|---|---|---|
| Garg A, Zhang X | 2019 | RNA-seq of the lens transition zone in control vs Pea3-depleted mouse embryonic tissue | https://www.ncbi.nlm.nih.gov/geo/query/acc.cgi?acc=GSE137215 | NCBI Gene Expression Omnibus, GSE137215 |

The following previously published dataset was used:

| Author(s) | Year | Dataset title | Dataset URL | Database and Identifier |
|---|---|---|---|---|
| Zhao Y, Zheng D, Cvekl A | 2019 | Profiling of chromatin accessibility and identification of general cis-regulatory mechanisms that control two ocular lens differentiation pathways | https://www.ncbi.nlm.nih.gov/geo/query/acc.cgi?acc=GSE124497 | NCBI Gene Expression Omnibus, GSE124497 |

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
