## [Decision Letter]

**Acceptance summary:**

This manuscript unraveled previously unappreciated aspects of lens development. More specifically, presented findings support the tenet that Etv 1, 4 and 5 play a role in lens development whereby underpinning mechanism may be at least in part distinct that those engaged by FGF.

**Decision letter after peer review:**

Thank you for submitting your article "Etv transcription factors functionally diverge from their upstream FGF signaling in lens development" for consideration by *eLife*. Your article has been reviewed by three peer reviewers, including Ivan Topisirovic as the Reviewing Editor and Reviewer #1, and the evaluation has been overseen by Kathryn Cheah as the Senior Editor.

The reviewers have discussed the reviews with one another and the Reviewing Editor has drafted this decision to help you prepare a revised submission.

This manuscript seeks to determine the role of Etv factors in lens development and establish the underpinning signaling mechanisms. Presented data support potential role of Etv 1, 4 and 5 in lens development, whereby the functional divergence between the latter factors and FGF is proposed. Notwithstanding that mouse genetics and the number of characterized phenotypes were found to be impressive, several important concerns were raised regarding interpretation of corresponding signaling perturbations.

Summary:

The focus of this article is to examine the roles of Etv1, Etv4, and Etv5 in lens development using Etv triple knockdown (Etv TKO) mice. Data are provided showing that Mek/Erk signaling dictates expression of Etv1, 4 and 5 in the transitional zone of the lens whereby Etv TKO mice exhibit alterations in lens development. Based on genome-wide steady-state mRNA analysis and subsequent genetic approaches, the authors suggest the model whereby downregulation of Spry2 leads to inhibition of Fgfr and Erk signaling. Moreover, in the Etv TKO background, Jagged1/Notch1 signaling is attenuated, while Tsc2 is downregulated leading to sustained mTORC1 activity. Overall, this manuscript is well written and of potential interest. Several issues were however observed. The major concern was that proposed signaling mechanisms were open to alternative interpretations and/or founded on data that were deemed to be too preliminary. It was concluded that these issues should be addressed to sufficiently support authors' conclusions.

Essential revisions:

1) Based on apparent complexity of the interplay between FGF and ERK signaling in Etv TKO lenses, several issues were raised considering data interpretation. To this end, although Etv loss resulted in Spry2 downregulation, it also decreased the expression of FGFR1 and 3. Considering that these events are expected to have an opposing effect on FGF signaling, it is important to examine how Fgfr1/Fgfr3 mRNA expression is spatially decreased in Etv TKO lens. Does decreased expression of Fgfr1 and 3 occur across the lens area or is it limited to the transitional zone? The authors should also discuss why fgfr1 and 3 reduction does not affect increase in phospho-ERK levels in Etv TKO. Finally, to corroborate claims regarding potential Etv-Fgf feedback, the authors should determine Spry2 levels in Erk-CKO and Mek-CKO lenses.

2) Better characterization of Etv TKO phenotypes is warranted. For example, the assumption that Fgfr loss result in same phenotypes as ablation of Mek and Erk was found to be inadequate, as pathways other than Mek/Erk also mediate the effects of Fgfr, while factors other than FGF can trigger Mek/Erk signaling. Accordingly, Fgfr mutants present with the smaller lens that appears as a vesicle with impaired primary fiber elongation, while the Mek/Erk mutants do not present this lumen. In addition, although Fgfr mutants cause effects similar to Sef overexpression in lens, they do not exactly mirror Spry2 mutants. It was also found that the statement that Etv mutants continue to express fiber-specific crystallin genes, while the Fgfr mutants do not, is not sufficiently substantiated by the data. The authors should therefore reconsider their conclusions as their data convincingly points on to the Erk1/2- Etv link, and not to direct connection to Fgfr signaling. Finally, considering the multitude of MAPKs, the authors should specify that they are monitoring Erk throughout the manuscript.

3) To further strengthen the role of Spry2 downregulation in Etv TKO lens phenotypes, the authors should also determine expression of Etv 1, 4 and 5 and Spry2 in Fgf3(OVE391) lenses. More specifically, the levels of Etv 1, 4 and 5 and Spry2 in anterior lens epithelium in Fgf3(OVE391) mice should be determined. Also, is Spry2 downregulated in anterior lens epithelium of Fgf3(OVE391)/*R26^EtvEnR^* mice? The authors should also acknowledge that Spry2 is not a 'specific' FGF signaling inhibitor, because it has been shown to suppress other receptor tyrosine kinases.

4) In experiments using *R26^EtvEnR^*, the authors presume that Etv functions only as a transcription activator. Monitoring the lens development in *R26^EtvVP16^* is an important control that is missing.

5) The authors implicate defects in Notch signaling in Etv TKO phenotypes based on an anterior shift of Foxe3/ Prox1. However, this shift is not that spatially obvious relative to the reduced size of the mutant lens, in particular as this is not accompanied by the corresponding shift in fiber crystallin patterns (Figure 4). Furthermore, it is difficult to determine whether the observed defects in Notch signaling are direct consequence of Etv loss, or caused indirectly via factors acting upstream. The authors should experimentally address this. Finally, the experiments in Figure 6 should be complemented by those monitoring levels of Notch1-ICD.

6) The mTOR part of the study was found to be underdeveloped. Considering complexity of feedbacks between Erk and mTOR signaling, the authors should use pharmacological (mTOR inhibitors) approaches to corroborate conclusions that Etv TKO phenotypes are driven by sustained mTOR activation.

7) Several inconsistencies and issue with data quality were observed. These include variable histology of Etv TKO lenses in different images (Figure 2) and peripheral sections through the eye, which may give misleading labeling [e.g. c-Maf in mutants (Figures 5O, P)]. In addition, size differences should not only be quantified for some of the mutants but also for Etv TKO mutants vs. controls. It was also found that it was difficult to appreciate the denucleation effects in Figure 7A, E and I. It is advised that these experiments are repeated using anti-lamin antibody, and calculating the number of nuclei followed by appropriate statistical analysis. Finally, there is apparent high variability of steady state mRNA levels between the replicates (Figure 2B). Herein, additional quality controls (e.g. showing how individual replicates and conditions segregate using PCA) should be included.

8) Description of certain procedures in Materials and methods section was found to be incomplete. Herein, the authors are encouraged to provide detailed description of how their EdU experiments including statistical analysis were carried out. Especially, in the histogram in Figure 1I where it was found that it is unclear what EdU-positive fraction represents. Is it the percentage of EdU positive cells in lens epithelium or percentage of total lens cells?

[Editors' note: further revisions were suggested prior to acceptance, as described below.]

Thank you for resubmitting your work entitled "Etv transcription factors functionally diverge from their upstream FGF signaling in lens development" for further consideration by *eLife*. Your revised article has been evaluated by Kathryn Cheah as the Senior Editor and Ivan Topisirovic as the Reviewing Editor.

The manuscript has been improved but there are some remaining issues that need to be addressed before acceptance, as outlined below:

1) Please refer to Figure 3—figure supplement 1 and Figure 5—figure supplement 1 in the Results section. These supplementary figures should be mentioned in an appropriate location of Results section.

2) Figure 3—figure supplement 1: It is shown that Spry2 mRNA expression is diminished in Erk CKO lens. Based on this, it was thought that the authors should refer to the Etv-FGF feedback scheme in the text.

3) Figure 5—figure supplement 1: It is advised that the authors discuss the Figure 5—figure supplement 1A-H after Figure 5A-N (subsection “Etv deficiency blocks aberrant but not normal lens differentiation”). In addition, the Figure 5—figure supplement 1C, F, I results should be elaborated after the first paragraph of the subsection “Deletion of Etv genes led to aberrant activation of mTOR signaling and disruption of disrupted nuclei clearance in the lens due to aberrant activation of mTOR signaling”.

4) Etv4 mRNA expression is abrogated in Fgf (OVE391); *R26^EtvENR^* lens, however, Etv1 and Etv5 mRNAs are still expressed. It is possible that Etv4 predominantly regulates its own transcription while having a lesser impact on Etv1 and Etv5 transcription. In this case, *R26^EtvENR^*-mediated Etv inhibition may be partially compared with Etv TKO. This possibility may explain why Spry2 expression is still retained in Fgf (OVE391); *R26^EtvENR^* lens (Figure 5—figure supplement 1L). The authors should consider discussing this.

5) Figure 1E, Figure 1J and Figure 3—figure supplement 1B are still of insufficient quality:

In Figure 1E, there is a pERK positive cell mass between the neural retina and dotted line. Can the authors comment as per what are these pERK-positive cells? Are they lens epithelial cells or ocular blood vessel?

In Figure 1J (Etv4 mRNA in situ) and Figure 3—figure supplement 1B (Fgfr1, Fgfr3 in situ) brown color noise signals still mask most of the lens fiber area, making it difficult to see blue signals. To corroborate author's conclusions, better quality images should be provided. In addition, the authors should use arrows to show in which subareas Fgfr1, 3 are expressed in the lens.

6) The description of mouse monoclonal anti-cyclin D1 in key resource table appears to be incomplete. This should be corrected.

---

## [Author Response]

Essential revisions:1) Based on apparent complexity of the interplay between FGF and ERK signaling in Etv TKO lenses, several issues were raised considering data interpretation. To this end, although Etv loss resulted in Spry2 downregulation, it also decreased the expression of FGFR1 and 3. Considering that these events are expected to have an opposing effect on FGF signaling, it is important to examine how Fgfr1/Fgfr3 mRNA expression is spatially decreased in Etv TKO lens. Does decreased expression of Fgfr1 and 3 occur across the lens area or is it limited to the transitional zone? The authors should also discuss why fgfr1 and 3 reduction does not affect increase in phospho-ERK levels in Etv TKO. Finally, to corroborate claims regarding potential Etv-Fgf feedback, the authors should determine Spry2 levels in Erk-CKO and Mek-CKO lenses.

We have performed RNA in situ hybridization to examine the expression pattern of Fgfr1/Fgfr3. As shown in Figure 3—figure supplement 1B, Fgfr1 and 3 expressions were reduced across the lens area in Etv TKO mutants, but the most prominent decrease occurred in the transitional zone. As we point out in the revised manuscript, it was previously reported that genetic deletion of both Fgfr1 and Fgfr3 did not affect lens growth and differentiation, as the intact Fgfr2 is apparently sufficient to support lens development (Zhao et al., 2008). This explains why the reduction in Fgfr1 and 3 expressions does not affect the increase in phospho-ERK levels in Etv TKO caused by the loss of Spry2. Lastly, we have added Figure 3—figure supplement 1C to show that Spry2 expression is indeed abolished in Erk-CKO lenses.

2) Better characterization of Etv TKO phenotypes is warranted. For example, the assumption that Fgfr loss result in same phenotypes as ablation of Mek and Erk was found to be inadequate, as pathways other than Mek/Erk also mediate the effects of Fgfr, while factors other than FGF can trigger Mek/Erk signaling. Accordingly, Fgfr mutants present with the smaller lens that appears as a vesicle with impaired primary fiber elongation, while the Mek/Erk mutants do not present this lumen. In addition, although Fgfr mutants cause effects similar to Sef overexpression in lens, they do not exactly mirror Spry2 mutants. It was also found that the statement that Etv mutants continue to express fiber-specific crystallin genes, while the Fgfr mutants do not, is not sufficiently substantiated by the data. The authors should therefore reconsider their conclusions as their data convincingly points on to the Erk1/2- Etv link, and not to direct connection to Fgfr signaling. Finally, considering the multitude of MAPKs, the authors should specify that they are monitoring Erk throughout the manuscript.

We agree with the reviewers that our current manuscript has provided strong evidence for the Erk1/2-Etv link, but the direct connection to Fgfr signaling is less well established. To demonstrate that both Erk and Etv are under control of FGF signaling, we have now examined the lens specific knockouts of Fgfr1/2. As shown in the new Figure 1A-H, genetic ablation of FGF receptors disrupted lens vesicle formation, confirming the critical role of FGF signaling in lens development. Importantly, both phospho-ERK and Etv1/4/5 expression are lost in Fgfr1/2 mutants. In addition, we have highlighted in the Discussion a previous study which showed that deletion of Fgf receptors after the lens vesicle stage disrupted phospho-ERK (Zhao et al., 2008). We have also cited our own work that deletion of FGF signaling mediators Frs2 and Shp2 abolished ERK phosphorylation and Etv expression in the lens (Pan et al., 2010 and Li et al., 2014). Altogether, we believe these evidence strongly support that FGF signaling regulates Erk and Etv activities in the lens. Lastly, we have replaced MAPK with ERK throughout the manuscript as the reviewer suggested.

3) To further strengthen the role of Spry2 downregulation in Etv TKO lens phenotypes, the authors should also determine expression of Etv 1, 4 and 5 and Spry2 in Fgf3 (OVE391) lenses. More specifically, the levels of Etv 1, 4 and 5 and Spry2 in anterior lens epithelium in Fgf3 (OVE391) mice should be determined. Also, is Spry2 downregulated in anterior lens epithelium of Fgf3 (OVE391)/R26^EtvEnR^ mice? The authors should also acknowledge that Spry2 is not a 'specific' FGF signaling inhibitor, because it has been shown to suppress other receptor tyrosine kinases.

We have performed RNA situ hybridization the reviewer requested. As expected, Etv1, 4, 5 and Spry2 are upregulated in the anterior lens epithelium in Fgf3(OVE391) mice, but their expressions are attenuated in Fgf3(OVE391)/*R26^EtvEnR^* mice (Figure 5—figure supplement 1). Following the reviewer’s suggestion, we have also revised the manuscript to refer Spry2 as “inhibitor of receptor tyrosine kinases”.

4) In experiments using R26^EtvEnR^, the authors presume that Etv functions only as a transcription activator. Monitoring the lens development in R26^EtvVP16^ is an important control that is missing.

Although *R26^EtvVP16^* would be a nice control, to our knowledge, this mouse line does not exist. Since it would take considerable time and effort to generate a new mouse line, this control experiment is unfortunately out of scope of the current study. We would like to note that Etv/1/4/5 proteins share a transcriptional activator domain (TAD), suggesting that they most likely behave as transcription activators. Indeed, both in vitro and in vivo studies have confirmed that *R26^EtvEnR^* blocks functions of Etv/1/4/5 (Curr. Biol. 11, 1739-1748, and Mao et al., 2009). Therefore, we are reasonably confident that *R26^EtvEnR^* is an effective tool to probe Etv functions in lens development.

5) The authors implicate defects in Notch signaling in Etv TKO phenotypes based on an anterior shift of Foxe3/ Prox1. However, this shift is not that spatially obvious relative to the reduced size of the mutant lens, in particular as this is not accompanied by the corresponding shift in fiber crystallin patterns (Figure 4). Furthermore, it is difficult to determine whether the observed defects in Notch signaling are direct consequence of Etv loss, or caused indirectly via factors acting upstream. The authors should experimentally address this. Finally, the experiments in Figure 6 should be complemented by those monitoring levels of Notch1-ICD.

We thank the reviewer for pointing out the subtle difference between the anterior shift of Foxe3/Prox1 expressions and the relatively normal fiber crystallin patterns. As is now elaborated in the Discussion, we believe that these phenotypes represent the accelerated differentiation of lens progenitor cells controlled by transcription factors Foxe3/Prox1 versus the slightly delayed terminal differentiation of lens fiber cells characterized by crystallin expression. As demonstrated by NICD staining (Figure 6D), Notch signaling is only active in lens progenitor cells, not in fiber cells. As such, it is not surprising that dysregulated Notch signaling in Etv TKO mutants cause the anterior shift of the transitional zone shown by Foxe3/Prox1 staining, but it is not expected to directly affect the fiber cell compartment. On the other hand, the terminal differentiation of the fiber cells may be slightly delayed in Etv TKO mutants, which compensates for the early onset of the lens progenitor cell differentiation. Ultimately, the lens fiber cells proceed to express the crystallin genes at the comparable levels as the wild type controls.

At the reviewer’s request, we have repeated the western blot experiment in Figure 6 with the additional control of monitoring Notch1-ICD levels. It confirmed that inhibition of Erk activity disrupted Jag1-Notch signaling in lens cells (Figure 6—figure supplement 1). In addition, we have showed that expression of Notch ligand Jag1 is compromised in Etv TKO mutants and our ChIP experiment demonstrated that Etv binds the Jag1 genomic locus. Although we cannot rule out an indirect role of Erk-Etv in regulating Notch signaling, we believe our results strongly support that the observed defects in Notch signaling are the direct consequence of Etv loss.

6) The mTOR part of the study was found to be underdeveloped. Considering complexity of feedbacks between Erk and mTOR signaling, the authors should use pharmacological (mTOR inhibitors) approaches to corroborate conclusions that Etv TKO phenotypes are driven by sustained mTOR activation.

We have attempted to rescue Etv TKO phenotype using the mTOR inhibitor Torin. However, intraperitoneal injection of Torin into the pregnant females failed to disrupt phospho-mTOR and phospho-4EBP staining in the lens of the wild type embryos, suggesting that mTOR signaling was not significantly altered by this approach. This could be due to the inefficient delivery of mTOR inhibitor into the lens via systemic injection. Since we are unable to perform this pharmacological experiment, we have toned down the link between Etv TKO nuclei clearance phenotype and mTOR, changing the subtitle of mTOR section to “Deletion of Etv genes led to aberrant activation of mTOR signaling and disruption of nuclei clearance in the lens”.

7) Several inconsistencies and issue with data quality were observed. These include variable histology of Etv TKO lenses in different images (Figure 2) and peripheral sections through the eye, which may give misleading labeling [e.g. c-Maf in mutants (Figures 5O, P)]. In addition, size differences should not only be quantified for some of the mutants but also for Etv TKO mutants vs. controls. It was also found that it was difficult to appreciate the denucleation effects in Figures 7A, E and I. It is advised that these experiments are repeated using anti-lamin antibody, and calculating the number of nuclei followed by appropriate statistical analysis. Finally, there is apparent high variability of steady state mRNA levels between the replicates (Figure 2B). Herein, additional quality controls (e.g. showing how individual replicates and conditions segregate using PCA) should be included.

We have replaced Figure 2A and E with better histological images and Figure 5O and P with central sections. The lens size difference between control and Etv TKO is now quantified in Figure 1I. As presented in Figure 7—figure supplement 1, we performed additional staining of the lens nuclei using both DAPI and anti-Lamin A/C antibody, and quantified the number of nuclei followed by One-way ANOVA test. Finally, we also presented PCA plot in Figure 3—figure supplement 1 to show the segregation of control and Etv TKO mutant RNA sequencing data.

8) Description of certain procedures in Materials and methods section was found to be incomplete. Herein, the authors are encouraged to provide detailed description of how their EdU experiments including statistical analysis were carried out. Especially, in the histogram in Figure 1I where it was found that it is unclear what EdU-positive fraction represents. Is it the percentage of EdU positive cells in lens epithelium or percentage of total lens cells?

The details for EdU experiments and the associated statistical analysis are now given in the Materials and methods section. Specifically, we clarify that the EdU-positive fractions represent the ratios of EdU-positive cells versus the total numbers of DAPI-positive cells within the lens epithelium.

[Editors' note: further revisions were suggested prior to acceptance, as described below.]

The manuscript has been improved but there are some remaining issues that need to be addressed before acceptance, as outlined below:1) Please refer to Figure 3—figure supplement 1 and Figure 5—figure supplement 1 in the Results section. These supplementary figures should be mentioned in an appropriate location of Results section.

Both supplementary figures are now mentioned in the Results section.

2) Figure 3—figure supplement 1: It is shown that Spry2 mRNA expression is diminished in Erk CKO lens. Based on this, it was thought that the authors should refer to the Etv-FGF feedback scheme in the text.

This is a good point. The Etv-FGF/Erk feedback scheme is now discussed in the last paragraph of the subsection “Genetic ablation of Etv causes ectopic activation of ERK signaling”.

3) Figure 5—figure supplement 1: It is advised that the authors discuss the Figure 5—figure supplement 1A-H after Figure 5A-N (subsection “Etv deficiency blocks aberrant but not normal lens differentiation”). In addition, the Figure 5—figure supplement 1C, F, I results should be elaborated after the first paragraph of the subsection “Deletion of Etv genes led to aberrant activation of mTOR signaling and disruption of disrupted nuclei clearance in the lens due to aberrant activation of mTOR signaling”.

These supplements are now described in the last paragraph of the subsection “Etv deficiency blocks aberrant but not normal lens differentiation”.

4) Etv4 mRNA expression is abrogated in Fgf(OVE391); R26^EtvENR^ lens, however, Etv1 and Etv5 mRNAs are still expressed. It is possible that Etv4 predominantly regulates its own transcription while having a lesser impact on Etv1 and Etv5 transcription. In this case, R26^EtvENR^-mediated Etv inhibition may be partially compared with Etv TKO. This possibility may explain why Spry2 expression is still retained in Fgf(OVE391); R26^EtvENR^ lens (Figure 5—figure supplement 1L). The authors should consider discussing this.

We have now discussed the partial inhibition of Etv by *R26^EtvEnR^* in the last paragraph of the subsection “Etv deficiency blocks aberrant but not normal lens differentiation”.

5) Figure 1E, Figure 1J and Figure 3—figure supplement 1B are still of insufficient quality:In Figure 1E, there is a pERK positive cell mass between the neural retina and dotted line. Can the authors comment as per what are these pERK-positive cells? Are they lens epithelial cells or ocular blood vessel?In Figure 1J (Etv4 mRNA in situ) and Figure 3—figure supplement 1B (Fgfr1, Fgfr3 in situ) brown color noise signals still mask most of the lens fiber area, making it difficult to see blue signals. To corroborate author's conclusions, better quality images should be provided. In addition, the authors should use arrows to show in which subareas Fgfr1, 3 are expressed in the lens.

The pERK positive cell mass under the lens ectoderm in the original Figure 1E were indeed ocular blood cells, which usually produce non-specific staining. To avoid confusion, we have replaced the image with a new one that does not have these blood cells. We also replaced Figure 1J and Figure 3—figure supplement 1B with better images that do not have the brown color noise and added arrows to indicate that *Fgfr1* and 3 are expressed in the transitional zone of the lens.

6) The description of mouse monoclonal anti-cyclin D1 in Key Resource Table appears to be incomplete. This should be corrected.

The information for Cyclin D1 antibody is now corrected in the Key Resource Table.